



# Towards a compound event-oriented climate model evaluation: A decomposition of the underlying biases in multivariate fire and heat stress hazards

Roberto Villalobos-Herrera[1,2], Emanuele Bevacqua[3], Andreia F.S. Ribeiro[4], Graeme Auld[5], Laura Crocetti[6,7], Bilyana Mircheva[8], Minh Ha[9], Jakob Zscheischler[10,11,12], and Carlo De Michele[13]

[1] School of Engineering, Newcastle University, Newcastle upon Tyne, NE2 1HA, United Kingdom

[2] Escuela de Ingeniería Civil, Universidad de Costa Rica, Montes de Oca, San José 1150-2060, Costa Rica

[3] Department of Meteorology, University of Reading, Reading, United Kingdom.

[4] Instituto Dom Luiz (IDL), Faculdade de Ciências, Universidade de Lisboa, 1749-016 Lisboa, Portugal.

[5] School of Mathematics, The University of Edinburgh, Edinburgh, United Kingdom.

[6] Department of Geodesy and Geoinformation, TU Wien, Vienna, Austria.

[7] Institute of Geodesy and Photogrammetry, ETH Zurich, Zurich, Switzerland

[8] Department of Meteorology and Geophysics, Sofia University, Sofia, Bulgaria.

[9] Laboratoire Atmosphères, Milieux, Observations Spatiales (LATMOS), Sorbonne Université, Paris and Guyancourt, France

[10] Climate and Environmental Physics, University of Bern, Bern, Switzerland.

[11] Oeschger Centre for Climate Change Research, University of Bern, Bern, Switzerland.

[12] Department of Computational Hydrosystems, Helmholtz Centre for Environmental Research - UFZ, Leipzig, Germany.

[13] Department of Civil and Environmental Engineering, Politecnico di Milano, Milan, Italy.

*Correspondence to*: Roberto Villalobos-Herrera (r.villalobos-herrera2@newcastle.ac.uk)

**Abstract.** Climate models' outputs are affected by biases that need to be detected and adjusted to model climate impacts. Many climate hazards and climate-related impacts are associated with the interaction between multiple drivers, i.e. by compound events. So far climate model biases are typically assessed based on the hazard of interest, and it is unclear how much a potential bias in the dependence of the hazard drivers contributes to the overall bias and how the biases in the drivers

interact. Here, based on copula theory, we develop a multivariate bias assessment framework, which allows for disentangling the biases in hazard indicators in terms of the underlying univariate drivers and their statistical dependence. Based on this framework, we dissect biases in fire and heat stress hazards in a suite of global climate models by considering two simplified hazard indicators, the wet-bulb globe temperature (*WBGT*) and the Chandler Burning Index (*CBI*). Both indices solely rely on temperature and relative humidity. The spatial pattern of the hazards indicators is well represented by climate models.

However, substantial biases exist in the representation of extreme conditions, especially in the *CBI* (spatial average of absolute bias: 21°C) due to the biases driven by relative humidity (20°C). Biases in *WBGT* (1.1°C) are small compared to the biases



driven by temperature (1.9°C) and relative humidity (1.4°C), as the two biases compensate each other. In many regions, also biases related to the statistical dependence (0.85°C) are important for *WBGT*, which indicates that well-designed physically-based multivariate bias adjustment should be considered for hazards and impacts that depend on multiple drivers. The proposed

compound event-oriented evaluation of climate model biases is easily applicable to other hazard types. Furthermore, it can contribute to improved present and future risk assessments through increasing our understanding of the biases' sources in the simulation of climate impacts.

## 1 Introduction

Understanding and assessing the risk of high-impact events induced by the combination of multiple climate drivers and/or

hazards, referred to as compound events, is challenging (e.g., Bevacqua et al., 2017; Manning et al., 2018; Zscheischler et al., 2020a). One of the reasons is that many high-impact events are caused by multiple variables that may not be extreme themselves, but their combination leads to an extreme impact (Zscheischler et al., 2018). For example, the risks associated with combined high temperature and high/low relative humidity such as heat stress and fires, can manifest in heat-related human fatalities (Raymond et al., 2020) and fire-induced tree mortality (Brando et al., 2014) even if the two contributing

variables are not necessarily extreme in a statistical sense. In the future, combinations of climate variables leading to disproportionate impacts will be affected by global warming, and reliable risk assessments are required (Fischer and Knutti, 2013; Russo et al., 2017; Schär, 2016; Raymond et al., 2020; Jézéquel et al., 2020; Zscheischler et al., 2020a). Therefore, a better understanding of how climate models represent the joint behaviour of variables behind compound events, such as temperature and relative humidity, is crucial to correctly quantify their associated hazards today and in the future (Zscheischler

et al., 2018).

Typically, the raw climate model data contains biases, which lead to biased estimates of climate risks (Maraun et al., 2017). Evaluating, i.e. assessing and understanding such biases is a crucial step towards impact modelling and thus assessment of future climate risks. Climate model evaluation is very often univariate, i.e. does not take into account the multivariate nature of many hazards that are driven by the interplay of multiple contributing variables (Vezzoli et al., 2017; Zscheischler et al.,

2018, 2019; Francois et al., 2020). However, evaluating the model representation of the individual contributing variables individually, and hence disregarding both the biases in the dependence between the contributing variables and how the biases in the drivers interact and influence the hazard, cannot provide direct information regarding the biases in the resulting hazard indicator. Furthermore, evaluating the hazard indicator only, e.g. heat stress regardless of the contributing variables temperature and relative humidity, may hide compensating biases in the contributing variables, even if the hazard indicator

appears to be well represented. An evaluation of climate models that considers the underlying multivariate nature of the hazards can provide a better physical understanding of the relevant model skills. In turn, a better understanding of model skills can serve as a basis for better adjustment of the biases and/or selection of best performing models, which are crucial for hazard assessment both in the present and future climate. However, studies evaluating the climate model multivariate representation





of hazard indicator are still rare (Bevacqua et al., 2019; Zscheischler et al., 2020b) and little is known on the effects of those

biases on multivariate hazards (Fischer and Knutti, 2013; Zscheischler et al., 2018).

In this study we propose a copula-based multivariate bias-assessment framework, which allows for decomposing the sources of bias in hazard indicators. We employ global climate model outputs from the fifth phase of the Coupled Model Intercomparison Project (CMIP5) and consider two simplified hazard indices, the Chandler burning index (*CBI*) for fire hazard and the wet bulb globe temperature (*WBGT*) index for heat stress, both driven solely by temperature and relative humidity.

Figure 1 illustrates the main rationale of the multivariate bias-assessment framework. Both hazard indices, *CBI* and *WBGT*, are influenced by the bivariate distribution of temperature and relative humidity (Figure 1c). Based on copula-theory, such a bivariate distribution can be decomposed in terms of the marginal distributions of temperature and relative humidity (Figure 1a, d), and their statistical dependence (Figure 1b). Hence, such a copula-based decomposition allows for understanding the biases in the hazard estimates in terms of both the contribution from the marginal distributions individually (Figure 1a,d; see

difference between grey and black lines), and from their statistical dependence (Figure 1b) (Vezzoli et al. 2017; Bevacqua et al., 2019). We present a methodology to quantify the role played by the biases in temperature, relative humidity, and their dependence, to the final bias in the fire and heat stress indices as simulated by climate models.

## 2 Data

### 2.1 Pre-processing

We employ 6-hourly data of 2-meter air Temperature (T) and Relative Humidity (RH) during the period 1979-2005 from ERA-Interim reanalysis (Berrisford et al., 2011; Dee et al. 2011) and twelve models from the CMIP5 multi-model ensemble (Taylor et al., 2012): ACCESS1-0, ACCESS1-3, bcc-csm1-1-m, BNU-ESM, CNRM-CM5, GFDL-ESM2G, GFDL-ESM2M, inmcm4, IPSL-CM5A-LR, NorESM1-M, GFDL-CM3 and IPSL-CM5A-MR; leap days were removed. To allow for an intermodel comparison, data were bilinearly interpolated to a 2.5° by 2.5° regular latitude-longitude grid. All oceanic grid cells

as well as those beneath 60°S were removed from all analysis, given that arguably no heat stress and fire risk exists in these areas.

Following Zscheischler et al. (2019), we restrict our analysis to the hottest calendar month of the year, which is selected based on the climatology of ERA-Interim data at each grid point. This choice was made because arguably heat stress and fire hazards tend to be more frequent during the warmest period of the year, and it avoids dealing with seasonality. Finally, for each model

and location, we consider the *T* and *RH* values at the daily 6-hourly time steps corresponding to the daily maximum temperature within the hottest month. The above results in a time series for each location and model, with daily values of the pair (T, RH). The resulting time series data are autocorrelated, which can compromise the interpretation of the statistical tests that we apply in the analysis (Yue et al., 2002; Dale and Fortin, 2009). Therefore, we carry out the analysis on the de-correlated time series, which are obtained from the original through subsampling every N=9 days, this is the minimum lag required to remove the

autocorrelation in *T* and *RH* time series data (at 95% confidence level), calculated for all grid points in ERA-Interim and the





CMIP5 models. Finally, all time series are sampled with the frequency of N. This is done N-times using different start epochs, where the first sampled time series starts with time epoch one, the second sampled time series with time epoch two and so on up to nine. The de-correlated time series of *T* and *RH* will henceforth be simply referred to as samples in the following sections. In the Appendix, Figure A1 illustrates, for a representative location in Brazil, one of the nine resulting samples of *T* and *RH*
for ERA-Interim and for a selection of CMIP5 models. The figure also shows how the bivariate interaction of these variables drive the fire and heat stress indices (coloured isolines) introduced in the next section.

## 2.2 Fire hazard and heat stress indices

We quantify fire and heat stress hazards based on two indices, i.e. *CBI* and *WBGT*, respectively. While more advanced and sophisticated indices exist for both of these hazards (e.g. Van Wagner, 1987; Fiala et al., 2011), here we employ these two
simplified indices. Our aim is to provide a methodological framework for a compound event-oriented evaluation of hazard indicators. Hence, employing simplified indices allows for the development of a test case of the methodological framework. We do not aim at providing an accurate assessment of the hazard; nevertheless, our results will provide indications that can serve as a basis for follow-up studies of more complex fire and heat stress hazards.

The CBI index was employed, for example, for studying fire risk in the United States (McCutchan and Main, 1989) and
globally (Roads et al., 2008). The index is based on air *T* (°C) and *RH* (%):

$$CBI = \frac{((110 - 1.73 * RH) - 0.54 * (10.20 - T)) * 1.24 * 10^{-0.0142 * RH}}{60} \qquad (1)$$

The 'simplified *WBGT*' (from now on *WBGT* for sake of brevity) index was developed by the American College of Sports Medicine (ACSM, 1984) as an indicator of heat stress for average daytime conditions outdoors. The index is defined as:

$$WBGT = 0.56T + 0.393e + 3.94 \qquad (2)$$

where $e = (RH/100) * 6.105 e^{(17.27T/237.7+T)}$ is water vapour pressure (expressed in hPa), which depends on air temperature and relative humidity.

## 3 Methods

This section presents the conceptual framework and a bias decomposition methodology used to analyse the multivariate indices described above. We then present an overview of the data processing before detailing the conventional statistical tests we have
incorporated into our test suite.

## 3.1 Copula-based conceptual framework

As both *CBI* and *WBGT* are functions of *T* and RH, it follows that their distributions are determined by the joint distribution of *T* and RH. Copula theory provides us with a natural way to decompose the joint distribution of *T* and *RH* in terms of the marginal distributions of *T* and *RH* (the distributions of the individual variables considered in isolation) and a term, known as



the copula, that describes the dependence between *T* and *RH* (Figure 1). This allows us to understand how bias in each of these components contributes to the bias in *CBI* and *WBGT*.

A copula is a function that completely characterizes the dependence structure between random variables, in our case *T* and RH. Sklar's theorem (Sklar, 1959) is a fundamental result in copula theory, which states that the joint distribution of the random variables is determined by the marginal distributions and their copula. Mathematically, in our bivariate case, given the

two variables *T* and RH, with marginal cumulative distribution functions (CDFs) $F_T$ and $F_{RH}$, and marginal probability density functions (pdfs) $f_T$ and $f_{RH}$, following Sklar's theorem, the joint pdf $f_{T, RH}$ can be decomposed as:

$$f_{T,RH}(T, RH) = f_T(T) \cdot f_{RH}(RH) \cdot c(U_T, U_{RH}) \qquad (3)$$

where $U_{RH} = F_{RH}(RH)$, $U_T = F_T(T)$, (note that *U* indicates that both $U_{RH}$ and $U_T$ are uniformly distributed by construction on the domain [0,1]), and *c* is the copula density, which describes the dependence of the joint distribution $f_{T, RH}$ independently

from the marginal distributions $f_T$ and $f_{RH}$. Note that eq. (3) naturally extends to the case of an arbitrary number of random variables (Bevacqua et al., 2017), however here we focus on the bivariate case. Copulas allow for great flexibility in modelling complex dependence structures between several variables and there are a huge variety of parametric copula families available for statistical modelling purposes (Nelson, 2006; Salvadori and De Michele, 2007; Salvadori et al., 2007; Durante and Sempi, 2015; Bevacqua et al., 2020a). However, note that following the methodologies developed by Rémillard and Scaillet (2009)

and Vezzoli et al. (2017), here we will consider a non-parametric framework, i.e. we will consider empirical, rather than parametric, distributions within our testing procedures. This choice avoids unnecessary parametric-based assumptions on the distributions that could bias the results about both univariate and multivariate features.

A characteristic of copulas is the invariance property (Salvadori et al. 2007, proposition 3.2), i.e. if $g_1$ and $g_2$ are monotonic (increasing) functions, then the transformed variables $g_1(T)$ and $g_2(RH)$ have the same copula as *T* and RH. This property is

crucial to the methodology described in the following section, where the monotonic functions are the marginal CDFs of *T* and *RH* (or their inverses).

## 3.2 Contribution of the bias in the drivers to the bias of *CBI* and *WBGT*

We assess how biases in each of the marginal distributions of $T_{mod}$ and $RH_{mod}$, and $C_{mod}$ (the copula of $T_{mod}$ and $RH_{mod}$) contribute to the bias in the representation of the extreme values of *CBI* and *WBGT* using their 95th percentile (Q95). This is

achieved based on a methodology originally introduced by Bevacqua et al. (2019) to attribute changes in compound flooding to its underlying drivers (and employed by e.g., Manning et al. (2019) and Bevacqua et al. (2020b)).

We carried out three experiments. In experiment i, we obtained, via a data transformation, a bivariate pair ($T_i$, $RH_i$) with copula $C_i$ where one component of the three underlying distributions ($T_i$, $RH_i$, $C_i$) is the same as that of a given CMIP5 model, and the other two components are the same as ERA-Interim. We then perform the quantile tests described in 3.3.3 for *CBI* (or

*WBGT*) using values based on ($T_i$, $RH_i$) and ($T_{erai}$, $RH_{erai}$). The specific experiments carried out are described below.





Experiment a), assessing the bias contribution of $T_{mod}$: From the variable $T_{erai}$ we calculated the associated empirical cumulative distribution $U_{T,erai}$. From the variable $T_{mod}$ we calculated the empirical CDF $F_{T,mod}$, through which we defined $T_a = F^{-1}_{T,mod}(U_{T,erai})$. The variable $T_a$ has the same distribution as $T_{mod}$ while the pair ($T_a$, $RH_{erai}$) has the copula of ERA-Interim.

Experiment b), assessing the bias contribution of $RH_{mod}$: This experiment follows the same structure as experiment (a) but with the roles of $T_{erai}$ and $RH_{erai}$ reversed, from which we get the pairs ($T_{erai}$, $RH_b$)

Experiment c), assessing the bias contribution of $C_{mod}$: From the variables ($T_{mod}$, $RH_{mod}$) we calculated the associated marginal empirical cumulative distributions ($U_{T,mod}$, $U_{RH,mod}$). From the variables $T_{erai}$ and $RH_{erai}$, we defined the empirical CDFs $F_{T,erai}$ and $F_{RH,erai}$, through which we defined $T_c = F^{-1}_{T,erai}(U_{T,mod})$ and $RH_c = F^{-1}_{RH,erai}(U_{RH,mod})$. The pair of variables ($T_c$, $RH_c$) have the same marginal distributions as the pair ($T_{erai}$, $RH_{erai}$) but the copula of the model, i.e., $C_{mod}$, since ($T_c$, $RH_c$) was obtained from ($T_{mod}$, $RH_{mod}$) by monotonic transformations of the margins.

### 3.3 Description of the testing procedure

The full data processing procedure is shown in Figure 2. We began with the ERA-Interim and CMIP5 data and obtained $T$ and $RH$ samples (see section 2.1). The CMIP5 samples were then subject to the transformation procedure described in section 3.2.4. This results in five sets of $T$ and $RH$ data corresponding to: the ERA-Interim reference, the original model sample, and the three experiments (a, b, and c) used to assess the bias contributions of biases in CMIP5 model T, $RH$ and their copula. At this stage, the $CBI$ and $WBGT$ indices are calculated on all five sets.

We execute univariate and multivariate non-parametric statistical tests to evaluate the properties of $CBI$, $WBGT$ and their driver variables (i.e., T, RH, and their dependence) prior to proceeding with our bias decomposition approach. Details for each of the tests are provided below but, in general, we follow a non-parametric approach similar to Vezzoli et al. (2017). Each of the nine de-correlated ERA-Interim samples was independently tested on a cell-by-cell basis against a different CMIP5 model sample, therefore each statistical test is repeated nine times per model. To adjust for multiple testing, we use the conservative Bonferroni correction method, which penalises the significance level α using the number of repeated tests m=9, so that the individual hypothesis tests are evaluated at an α/m significance level (Jafari and Ansari-Pour, 2019). A 5% significance level is used, which after Bonferroni correction was adjusted to 0.0056 for use in each individual hypothesis test. All of our analysis was carried out in R (R Core Team, 2019), and the functions used for each test are detailed in their corresponding section below.

Graphically, the statistical test results are presented as a percentage of all 108 CMIP5 model samples (9 samples times 12 models) where the null hypothesis is rejected. The percentage we consider is calculated at each grid cell, and stippling is added where the null hypothesis is rejected in at least 75% (81/108) of all CMIP5 model samples.





### 3.3.1 Univariate evaluation of T, RH, *CBI*, and *WBGT*

In order to understand how faithfully the marginal distributions of T, RH, *CBI* and *WBGT* from the ERA-Interim data are represented in a given CMIP5 model, we perform the two-sample Anderson-Darling (AD) test via the ad.test function of the kSamples R-package (v1.2-9; Sholz and Zhu, 2019). This is a non-parametric procedure that considers the null hypothesis:

"the two samples are from the same distribution". AD was selected over the Kolmogorov-Smirnov test as it is more sensitive to differences in the tails of the two distributions, while there is also evidence that it is the more powerful among the two tests (Engmann and Cousineau, 2011).

### 3.3.2 Dependence between *T* and RH

A simple way to test how well the dependence between the variables *T* and *RH* in ERA-Interim is represented in a given

CMIP5 model, is to compare the calculated values of some statistical measure of association. Here we use Kendall's $\tau$ rank correlation. The cor.test function of the core stats R-package was used to perform all $\tau$ calculations (R Core Team, 2019).

To test whether the values of $\tau$ obtained from a given model sample differ in a statistically significant way from the corresponding ERA-Interim values. We begin by considering the approximate $100(1-\alpha)\%$ confidence interval $(\tau_L, \tau_U)$ for $\tau$ associated with the point estimator $\hat{\tau}$ given by:

$$\tau_L = \hat{\tau} - z_{\alpha/2}\hat{\sigma}, \tau_L = \hat{\tau} + z_{\alpha/2}\hat{\sigma} \qquad (4)$$

where $\hat{\sigma}^2$ is an estimator of $\text{var}(\hat{\tau})$ (Hollander et al., 2014). For our testing we calculate $\hat{\sigma}^2$, $\hat{\tau}$ and the confidence interval $(\tau_L, \tau_U)$ for each grid cell in all ERA-Interim samples, using a customised version of the kendall.ci function included in the NSM3 R-package (v1.15, Schneider et al., 2020). The CMIP5 model samples are then evaluated in two ways. Firstly, if the model sample value of $\tau$ lies within the confidence interval calculated for its corresponding ERA-Interim sample, the model

sample is judged to not significantly differ from ERA-Interim in terms of the rank correlation between *T* and RH. Secondly, we calculated the $z_{\alpha/2}$ and hence the $\alpha$ or p-value for each sample, these were tested for significance using the same Bonferroni-adjusted value of 0.0056 used in the univariate testing. The results from both testing methodologies are consistent with each other, we present the ad-hoc p-value test results in the main text and the confidence interval tests are included in the appendix.

Note that different copulas may give rise to the same value of $\tau$, therefore we cannot conclude that a model that faithfully reproduces the ERA-Interim values of $\tau$ is accurately representing the full dependence structure between *T* and RH. Therefore, we account for the dependence structure by also carrying out hypothesis tests which are based on the full copula function. We perform the non-parametric test of copula equality based on the Cramer-von-Mises test statistic proposed by Remillard and Scaillet (2009), used in Vezzoli et al. (2017) for testing the capability of a climate-hydrology model to reproduce the

dependence between temperature, precipitation and discharge for the Po river basin in Italy, and recently employed by Zscheischler and Fischer (2020) for evaluating the ability of climate models to represent the dependence between temperature





and precipitation in Germany. The copula equality test has a null hypothesis of $H_0$: $C_{erai} = C_{mod}$ where $C_{erai}$ and $C_{mod}$ are the copulas of *T* and *RH* represented in ERA-Interim and a given model respectively, with the alternative hypothesis being that these copulas differ. We used the TwoCop function of the TwoCop R-package (v1.0, Remillard and Plante, 2012) to run the test.

### 3.3.3 Bias in the representation of extreme events of *CBI* and *WBGT*

To evaluate how well CMIP5 models simulate extreme values of *CBI* and *WBGT*, we compare high quantiles (i.e. the 95th percentile Q95) of these indices from each model with those of ERA-Interim. To assess whether the observed differences in the quantiles are statistically significant, we calculate the 95% confidence intervals for the Q95 of *CBI* and *WBGT* at each location for ERA-Interim based on 1000 bootstrap samples. Like our evaluation of Kendall's τ, if the model index lies outside the confidence interval we consider the model has a significantly different representation of extreme values of *CBI* and *WBGT* from ERA-Interim.

## 4 Results

### 4.1 Univariate evaluation of T, RH, *CBI*, and *WBGT*

#### 4.1.1 *CBI* and *WBGT*

We began our analysis by visualising the multimodel mean of the mean values of *CBI* and *WBGT* during the hottest months. According to reanalysis, the mean *CBI* is highest in regions with dry and warm weather during the hottest month, such as the Sahara, most of Australia, and the western USA and Mexico (Figure 3a). In contrast, *CBI* tends to be low in humid and warm regions such as the Amazon and Congo basins. We move to evaluating the CMIP5 model biases in mean *CBI*, which appear large in magnitude (compare Figure 3b and 3a); most landmass covered in dark red or blue colours, indicating CMIP5 multimodel mean bias of over 10°C from the ERA-Interim. In addition, AD test results show that 59% of global land mass has significant differences between ERA-Interim and CMIP5 distributions of *CBI* in at least 75% of model samples. Despite such biases in the representation of the mean *CBI* magnitude, the overall spatial patterns in mean *CBI* are well reproduced by the models. In fact, the *area weighted pattern correlation* (Pfahl et al., 2017), from now on *pattern correlation*, between models and reanalysis of mean *CBI* is high for all CMIP5 models, with a minimum value of 0.77 for the BCC CSM1.1.M model (Figure A2a shows the multimodel mean of mean *CBI*).

In the reanalysis data, mean *WBGT* values over 30°C are reached over most tropical land masses during each location's hottest month, with lower values in higher latitudes and the highest values near the Equator (Figure 3c). For *WBGT*, the pattern correlation between models and ERA-Interim is higher than for *CBI*, with a minimum value of 0.89 for the BCC CSM1.1.M model (Figure A2b shows the multimodel mean of mean *WBGT*). Mean multimodel bias in *WBGT* shows large parts of the





continents are within the +/- 0.5°C range relative to ERA-Interim. The AD test results indicate that the *WBGT* distributions in CMIP models are typically better than those of *CBI*; only 35% of grid cells fail our performance criterion (Figure 3d).

Overall, CMIP5 models underperform in key regions associated with high fire and heat stress hazards. *CBI*'s distribution is

not well represented by most CMIP5 models in regions characterized by high fire hazard levels such as the western USA and the Mediterranean basin, while CMIP5 *WBGT* results are significantly different to reanalysis in regions of high heat stress such as the Indian subcontinent and equatorial Africa.

### 4.1.2 *T* and RH

Following the evaluation of *CBI* and *WBGT*, we move towards evaluating how CMIP5 models represent the driving variables

of the hazard indicators, i.e. T, RH, and their statistical dependence. We first confirm that the expected latitudinal variation in *T* is present in ERA-Interim reanalysis (Figure 4a), and that *RH* is low over known desertic areas (Figure 4c).

The spatial pattern of mean *T* is well represented by CMIP5 models (Figure A2c), with all models showing a pattern correlation over land with ERA-Interim above 0.93. However, significant differences in the representation of the distributions (based on the AD test) are found over the Amazon basin, where the multimodel mean bias in mean *T* is positive, and over Northern

Africa and the Middle East, where the bias in mean *T* is negative (Figure 4b). Overall, we found that the area weighted multimodel mean of absolute value of the *T* bias is 1.6°C. The AD test results show that CMIP5 models fail to reproduce the observed ERA-Interim distribution of *T* over 40% of global land mass.

We find worse model skills in representing the *RH* distribution; in fact, models failed the AD test over 59% of the global land mass (Figure 4d). The spatial pattern of *RH* is not as well represented as that of T, with minimum and maximum pattern

correlations of 0.75 and 0.90 respectively (Figure A2d). The mean multimodel bias in *RH* is particularly large in the Amazon basin. Nevertheless, there are areas where the bias is relatively small, e.g., in Australia, Sahara, and eastern Asia. Notably, there is a clear resemblance between the bias patterns of mean *RH* (Figure 4d) and *CBI* (Figure 3b), with regions with high positive bias in *RH* corresponding to regions with strong negative bias in *CBI*, and an identical percentage of land mass showing significant differences. No similar behaviour is found for *WBGT*, i.e. the *WBGT* bias spatial pattern is similar neither to that

of *T* nor *RH* bias. We will investigate this behaviour in *CBI* and *WBGT* in further detail in section 4.3.

### 4.2 Dependence between *T* and RH

The results for our tests on the dependency structure of *T* and *RH* in CMIP5 models are shown in Figure 5. Figures 5a and 5b show Kendall's $\tau$ correlation between *T* and *RH* based on ERA-Interim reanalysis and the mean multimodel bias of this correlation, respectively. *T* and *RH* are strongly negatively correlated (Figure 5a), with an area weighted mean value of -0.50

(virtually all landmass has a significant correlation; not shown - results based on the indepTest function of the copula R-package (v0.999-19.1; Hofert et al., 2018)). The presence of a negative correlation is illustrated in Figures 1 (and A1 for a representative location). The area weighted absolute mean multimodel bias in $\tau$ is 0.095. The bias in $\tau$ is not significant for most of the global landmass for most of the models, i.e., the modelled correlations lie within the 95% confidence interval of $\tau$





of ERA-Interim (see infrequent stippling over 5.3% and 9.3% of land masses in Figures 5b and A3 respectively). Results are
similar for the copula equality test (Figure 5c), with an over 80% agreement in copula structure between ERA-Interim and
models for 52% of landmasses, and 60-80% agreement in 33% of landmasses. Overall, the regions where we detect the highest
amount of statistically significant differences in the copula structure and τ include parts of the Horn of Africa, India, and
Amazon basin (see also Figure A1c and Figure A1d where the model values have different distributions to ERA-Interim).

## 4.3 Contribution of the bias in the drivers to the bias in *CBI* and *WBGT* extremes

### 4.3.1 Drivers of the biases in *CBI* extremes

We now assess the biases in the representation of extreme events (95th quantile, Q95) in the *CBI* index, and the associated
drivers of the biases (Figure 6). The spatial pattern of the CMIP5 multimodel mean of Q95 (Figure 6b) is very similar to that
of ERA-Interim (Figure 6a). Figure 6c shows the biases in extreme *CBI*, whose highest values are in South America, central
North America, and parts of central Asia; which is in line with the biases in mean *CBI* (Figure 3b). The area weighted mean
of absolute bias in CMIP5 model *CBI* is 21°C, which is large compared to the area weighted mean *CBI* in ERA-Interim of
84°C (i.e. corresponding to 25%). In fact, the stippling over 75% of land masses in Figure 6c indicates that the models differ
significantly from ERA-Interim.

The bias in *RH* is the main contributor to total mean bias in extreme *CBI* values (Figs. 6d-f). The relevance of *RH* for the bias
in *CBI* is visible from the similarities in magnitude and spatial distribution of bias between Figure 6c and 6e. Furthermore,
while the area weighted mean of absolute bias in *CBI* is 21°C, the corresponding mean biases due to T, RH, and the dependency
between them are 3°C, 20°C and 3°C respectively. The relevant contribution of *RH* to the *CBI* index bias is in consistent with
the definition of the index, which is mainly influenced by *RH* and to a lesser extent by *T* (see nearly horizontal *CBI* isolines in
Figure 1); hence, also the dependency between *T* and *RH* plays a negligible role. As a result, while *RH* bias contributions drive
significant biases in *CBI* about everywhere but in the Sahara and Australia (see stippling over 73% of land masses in Figure
6c), *T* and dependence do not drive significant biases in *CBI* (see near-complete absence of stippling in Figure 6d and 6f).

A closer examination of the bias decomposition results shows, for a site with large positive bias in Brazil, that the results
shown in the multimodal mean bias plots (Figure 6c, e) reflect intermodel model behaviour at the local level. That is, CMIP5
models with high *RH* bias contributions also show high overall *CBI* bias (Figure 7a). At this location, there is a positive
intermodel correlation between the biases driven by *T* and *RH* (τ = 0.82; Figure 7b). Such behaviour is due to the combination
of the following two reasons: (1) a negative intermodel correlation between the biases in *T* ad RH, i.e. CMIP5 models
simulating too high temperatures also tend to simulate too low relative humidity (as discussed by Fischer and Knutti (2013));
and (2) the fact that *CBI* is high for low *RH* and high T. This feature is discussed in more detail in section 5. Similar results to
those discussed above for the site in Brazil are also observed for another representative location in South Africa with large
negative bias in *CBI* (SI Figure A4a). These locations are indicated throughout map plots with X markers.



### 4.3.2 Drivers of the biases in *WBGT* extremes

The spatial pattern of Q95 in ERA-Interim (Figure 8a) and in the CMIP5 multimodel mean (Figure 8b) is similar with low values concentrated along mountain ranges such as the Andes and Himalayas and in high latitudes, and the highest values located in South America and the Indian subcontinent. In several regions worldwide, CMIP5 models tend to underestimate Q95 values of *WBGT* (global area weighted mean bias of -0.35°C) and show significant biases relative to ERA-Interim along the tropics and subtropics (Figure 8c). However, in terms of values of the bias, the CMIP5 representation of the *WBGT* appears better than that of *CBI*. The area weighted mean of absolute bias in the index is 1.1°C (Figure 8c), which is small compared to the area weighted mean *WBGT* in ERA-Interim, i.e. 29°C (Figure 8a).

The decomposition of the bias shows that unlike *CBI* there is no single dominating source of bias in extreme values of *WBGT* (Figure 8d-f); all three possible sources contribute to the overall bias. Importantly, a degree of compensating biases is evident when comparing the multimodel mean biases driven by *T* (Figure 8d) and *RH* (Figure 8e). Large biases of opposite signs are evident over South America, central Asia, and other landmasses; hence, in these areas, the resulting biases in *WBGT* tend to be small (Figure 8c). Significant but opposite biases in *T* and *RH* (see stippling in Figures 8d and 8e) result in nonsignificant biases in *WBGT* (Figure 8c) over regions such as North America's Mississippi basin and around Zaire in central Africa. Globally, this compensating behaviour can be observed in the percentages of land mass where each bias component is significant. *T* and *RH* driven biases are significant over 69% and 48% of global land mass respectively, while copula biases are significant over 12%; however, the total bias in *WBGT* is only significant over 39% of land masses. Further evidence for these compensating biases can be found by observing that the area weighted average of absolute bias in Q95 *WBGT*, i.e. 1.1°C, is smaller than the contributions from *T* and RH, i.e. 1.9°C and 1.4°C, respectively. In addition, we observe a tendency towards a lower bias, on average, driven by the copula component (global area weighted average of absolute bias equal to 0.85°C); note that, however, some relevant positive bias contributions exist over eastern Brazil and central Africa where the copula test shows higher frequencies of rejection (Figure 5c), and a negative contributions over northern Russia, Central United States, and eastern Europe (Figure 8f).

The compensating bias in *T* and *RH* found above is in line with the findings of Fischer and Knutti (2013). Their results indicate that, at the local scale and for individual models, the biases in *WBGT* driven by *T* and *RH* tend to cancel each other out, resulting in small biases in the heat stress index. We find that this behaviour in individual models is reflected in the multimodel mean result (Figure 8c) in regions where most models have similar behaviours, e.g. where most models show a positive *WBGT* bias contribution from *T* (Figure 8d) and a negative one from *RH* (Figure 8e). We confirm the behaviour in individual models for two representative locations. In Brazil, the small mean bias in *WBGT* Q95 for all CMIP5 models results from mostly positive and negative biases driven by *T* and RH, respectively, across models (Figure 9a; the figure also indicates that the bias driven by the dependence is small and positive). In particular, models affected by a positive *T* bias contribution in *WBGT* because of too high *T* tend also to be affected by a negative *RH* bias contribution because of too low *RH* (Figure 9b). The compensation of the biases in individual models arises from (1) opposite biases in *T* and *RH* (models simulating too high





temperatures also tend to simulate too low relative humidity; Fischer and Knutti (2013)), and (2) the *WBGT* tendency to be high (low) for humid and warm (dry and cold) conditions. Figure A5 illustrates such a cancellation of the bias in *WBGT* for a

location in South Africa, where the negative dependency between *T* and *RH* leads to a small bias in *WBGT*. In this location, the model biases driven by *T* are negative, therefore those driven by *RH* are positive.

## 5 Discussion

Based on copula theory we have designed a non-parametric procedure for multivariate bias assessment, which decomposes the bias in hazard indicators into the bias in the drivers and their dependence. We apply our approach to study the contribution

of T, RH, and their dependence to the bias in the fire and heat stress hazard indices *CBI* and *WBGT*. CMIP5 climate models show a good overall representation of the one-to-one pattern correlations of T, RH, *CBI* and *WBGT*, consistent with an acceptable representation of the first-order global-scale atmospheric circulation. However, considerable biases in *CBI* and *WBGT* over geographical areas with critical fire and heat stress risk exist.

We found that the role played by the bias in the drivers to the bias in the associated hazard indicator differs among the hazards

and at the regional level. While *CBI* biases are mainly driven by biases in RH, for *WBGT* biases the interplay between biases in T, *RH* and their dependence matters for a number of areas including eastern Brazil, Africa, and parts of central North America and India. The geographical areas where the bias in the copula between *T* and *RH* is more relevant to the bias in *WBGT* coincide with the regions where the copulas of most models tend to differ from those of the reanalysis.

These findings exemplify the need for multivariate bias adjustment methods, which can adjust climate model biases in the

dependencies between multiple drivers of hazards (Francois et al., 2020; Vrac, 2018). Climate model output should be a reliable input for the bias adjustment methods, e.g., models should provide a plausible representation of large-scale atmospheric circulation (Maraun et al., 2016; 2017). The relevance of multivariate bias adjustment methods is also supported by the fact that adjusting biases variable by variable may even increase biases in impact-relevant indicators (Zscheischler et al., 2019). Nevertheless, Zscheischler et al. (2019) found that univariate bias-adjustment is relatively efficient in the case of *CBI*, while

multivariate methods lead to much stronger reductions in the case of *WBGT*. This is consistent with our finding that the bias in *RH* is the main contributor to the bias in *CBI*, which is due to the fact that *CBI* variability is mainly driven by *RH* variability and to a lesser extent by *T* variability. It should be noticed, however, that the considered fire indicator *CBI* is overly simplistic. In practice, weather conditions that promote fires are also related to wind speed and previous rainfall, which are for instance included in the Forest Fire Weather Index (FWI, Van Wagner, 1987), as well as fuel availability and aridity.

The relative biases in *WBGT* extremes are smaller compared to the relative biases in *CBI* even though biases in *WBGT* are typically related to both biases in *T* and RH. In fact, biases in *WBGT* are smaller than the bias contributions from *T* and RH. This demonstrates the presence of compensating biases for *WBGT*. In line with Fischer and Knutti (2013), models which tend to simulate too high *T* also tend to simulate too low *RH* (and vice versa), which results in relatively smaller absolute biases in the *WBGT* of individual models. A negative intermodel correlation between the contributions of *T* and *RH* to *WBGT* biases





reduces the biases in *WBGT* in the CMIP5 average. For the fire hazard, one may expect an enhancement of the *T* and *RH* biases rather than a compensation (Fischer and Knutti, 2013), given that we have found a positive correlation between the bias driven by *T* and RH. However, we found that such a potential enhancement of the biases caused by the bivariate interaction between the bias driven by *T* and *RH* does not occur for the considered fire hazard indicator, i.e. *CBI*, because the index is mainly controlled by *RH* (see isolines in Figure 1c), which also controls the biases of the index. These results underline the importance

of attributing the sources of biases in hazard and risk indicators on a compound/multivariate perspective in terms of the dependency between the driver variables, rather than focusing on purely univariate assessments. The presented bias-decomposition method would potentially become even more relevant when considering more complex hazard indicators driven by more variables, such as the case of fire hazard as outlined above. This would require an extension of the bivariate copula framework, potentially using vine copula decompositions such as those employed in Hobæk Haff et al. (2015).

Given the critical importance of addressing compound/multivariate events that are often associated with extreme impacts (Leonard et al., 2014; Zscheischler et al., 2018), we assessed the bias decomposition for high quantiles of *CBI* and *WBGT*. The extremes of the considered indicators are not necessarily caused by extreme values of the drivers. Hence, the characterization of the dependence structure between their climate drivers (i.e., *T* and RH) was performed in terms of their full joint distribution to capture all the events, i.e. we did not only consider the combination of simultaneous *T* and *RH* extremes. However,

depending on the type of hazard considered, investigating biases in the tail dependence between the drivers may be relevant to understanding the biases in the hazard. For example, the tail dependence between storm surge and precipitation, which is relevant for compound coastal flooding, may be slightly underestimated in CMIP5 models (Bevacqua et al., 2019). Similarly, there is evidence that the tail dependence between hot and dry conditions may be underestimated by climate models in some cases (Zscheischler & Fischer, 2020).

The present methodology can be used for assessing the sources of bias in other types of compound events caused by other sets of dependent drivers, such as compound drought and heat (Zscheischler and Seneviratne, 2017) and compound coastal flooding (Bevacqua et al., 2020b). Other types of compound events, e.g., temporal clustering of storms (Bevacqua et al. 2020c; Priestley et al., 2017) and simultaneous extreme events in distant regions (Kornhuber et al., 2020), can also lead to large impacts and are therefore relevant for the impact community. A compound event-oriented evaluation of impacts similar to that

proposed here, i.e. disentangling the biases in the individual physical drivers, could be adopted in future studies to aid present and future impact assessments.

## 6 Conclusions

Climate model data contains biases that need to be evaluated and ultimately adjusted to avoid misleading risk assessments. However, while many climate-related extreme impacts are caused by the combination of multiple variables, i.e. compound

events, climate model evaluation methods typically do not consider the multivariate nature of the hazards. In this study, we took a compound event-perspective and, based on copula theory, introduced a multivariate bias-assessment framework, which



allows for disentangling and better understanding the multiple sources of biases in hazard indicators. Here, we investigated how the biases in temperature, relative humidity, and their dependence, affect the overall biases in fire and heat stress indicators (*CBI* and *WBGT*, respectively). We found that biases in *CBI* are mainly driven by biases in relative humidity, in line with the

fact that the index is only marginally affected by temperature. In contrast, the biases in *WBGT* are often driven by biases in temperature, relative humidity, and their statistical dependence. Opposing biases in temperature and relative humidity tend to compensate for each other, resulting in relatively small biases in *WBGT*. The results highlight areas where a careful interpretation of these indicators is required and where multivariate bias corrections of temperature and relative humidity should be considered future risk assessments.

Given the relevance of compound weather and climate events for societal impacts, the presented framework could be useful in further studies aiming at disentangling and better understanding the drivers of the biases in the representations of other impacts. The framework could also be useful to assess biases among drivers of hazards when data for the hazard indicators are not available. A compound event oriented model evaluation of modelled impacts and associated drivers would be beneficial for disaster risk reduction and, ultimately, could feedback into climate model development processes and stimulate the design

of new bias-adjustment methods.

**Author contributions**

Emanuele Bevacqua and Carlo De Michele conceived the study and supervised the project. Roberto Villalobos Herrera carried out the analysis of the biases based on the data prepared by Emanuele Bevacqua, Andreia F.S. Riberio, Laura Crocetti, Graeme Auld, Bilyana Mircheva, Minh Ha. Roberto Villalobos Herrera prepared all the figures, but Figure 1 and A1 were prepared by

Emanuele Bevacqua and Andreia Riberio. The paper was written by Graeme Auld, Emanuele Bevacqua, Carlo De Michele, Andreia Ribeiro, and Roberto Villalobos Herrera. Jakob Zscheischler contributed to the development of the idea of the work and helped with final edits. All the authors discussed the results of the paper.

**Acknowledgements**

This work emerged from the Training School on Statistical Modelling organized by the European COST Action DAMOCLES

(CA17109). We acknowledge the World Climate Research Programs Working Group on Coupled Modelling, which is responsible for CMIP, and we thank the climate modelling groups for producing and making available their model output. EB acknowledges financial support from the European Research Council grant ACRCC (project 339390) and from the DOCILE project (NERC grant: NE/P002099/1). AFSR acknowledges the Portuguese Foundation for Science and Technology (FCT) for the grant PD/BD/114481/2016 and the project IMPECAF (PTDC/CTA-CLI/28902/2017). CDM acknowledges the Italian

Ministry of University and Research through the PRIN2017 RELAID project. The work of LC was supported by the TU Wien Wissenschaftspreis 2015, awarded to Wouter Dorigo. RVH acknowledges support from the University of Costa Rica and the





Newcastle University School of Engineering. JZ acknowledges support from the Swiss National Science Foundation (Ambizione grant 179876) and the Helmholtz Initiative and Networking Fund (Young Investigator Group COMPOUNDX, Grant Agreement VH-NG-1537).

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

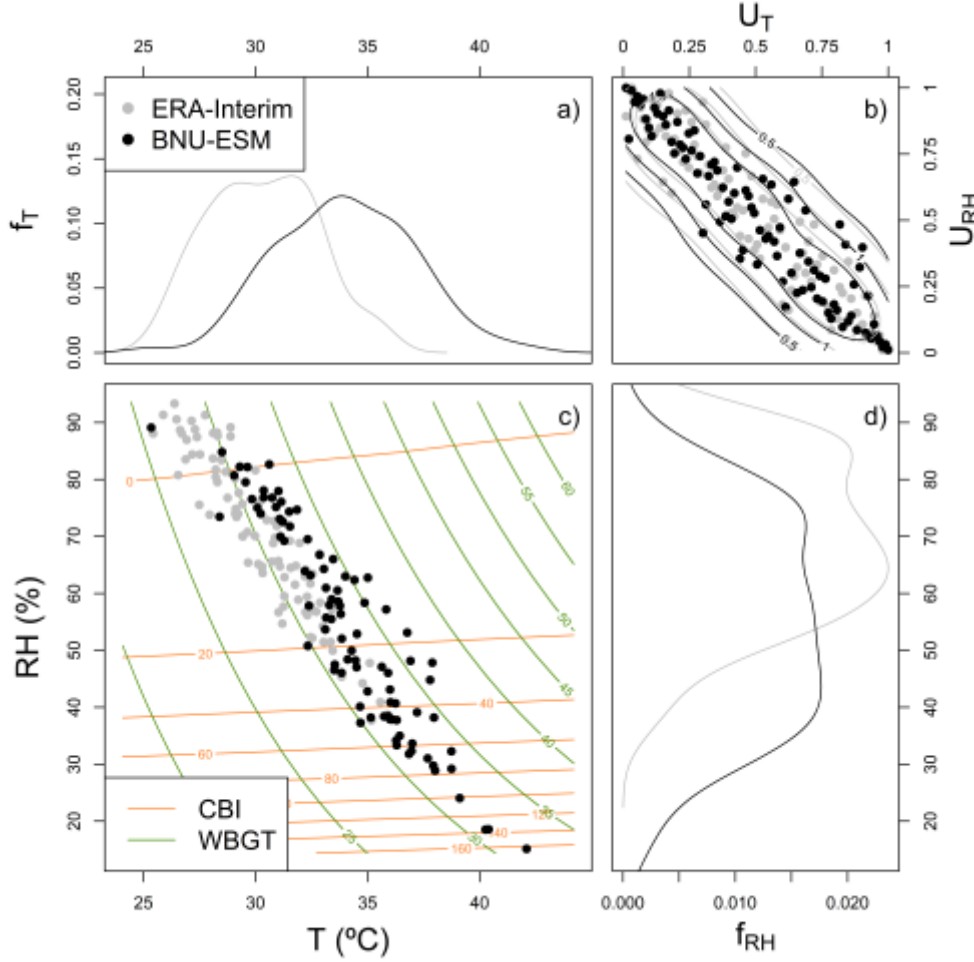

**Figure 1: Copula-based conceptual framework developed in this study to evaluate biases in *CBI* and *WBGT* indices. The framework is illustrated for a representative location in Brazil (Amazon, 5ºS and 56.5ºW; indicated via X markers in the next figures). Panel (c) shows that both *CBI* and *WBGT* indicators are functions of Temperature *T* and Relative humidity *RH* (see isolines of equal levels of *CBI* in orange and *WBGT* in green). Panel (c) also shows the bivariate distribution of (T, RH) within which grey and black dots show data for ERA-Interim and CMIP5 model BNU-ESM, respectively (period 1979-2005). Copula theory allows for decomposing the bivariate probability density function (pdf) of *T* and *RH* in terms of the marginal pdf of *T* (a) and *RH* (d) and the copula density that describes their dependence (b) (see the text for more details). Such a decomposition allows for disentangling the biases in the indices in terms of the driving biases in the marginals and copula; here, these biases are visible as the differences in the empirical pdf of the CMIP5 model (black) and the reference dataset, i.e. ERA-Interim (grey).**

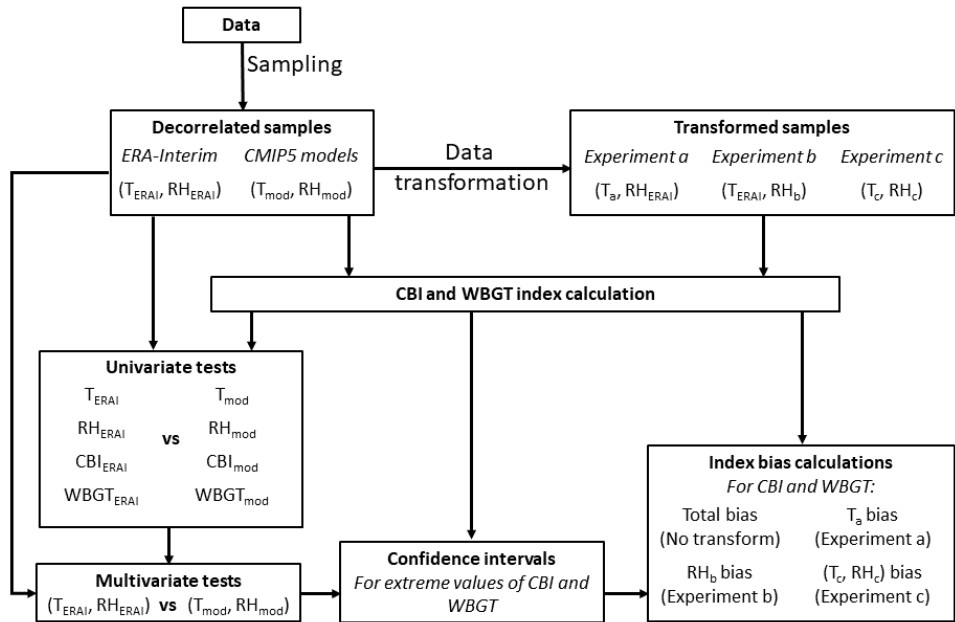

**Figure 2: Block-diagram showing the data and methods used. Temperature (T) and relative humidity (RH) decorrelated samples from CMIP5 models' biases are analysed using univariate and multivariate statistical tests using ERA-Interim as reference dataset. We also create transformed CMIP5 model samples, which allow for assessing the bias in the extreme values of the hazard indicator (*CBI* and *WBGT*) driven by biases in T, RH, as well as their statistical dependence.**


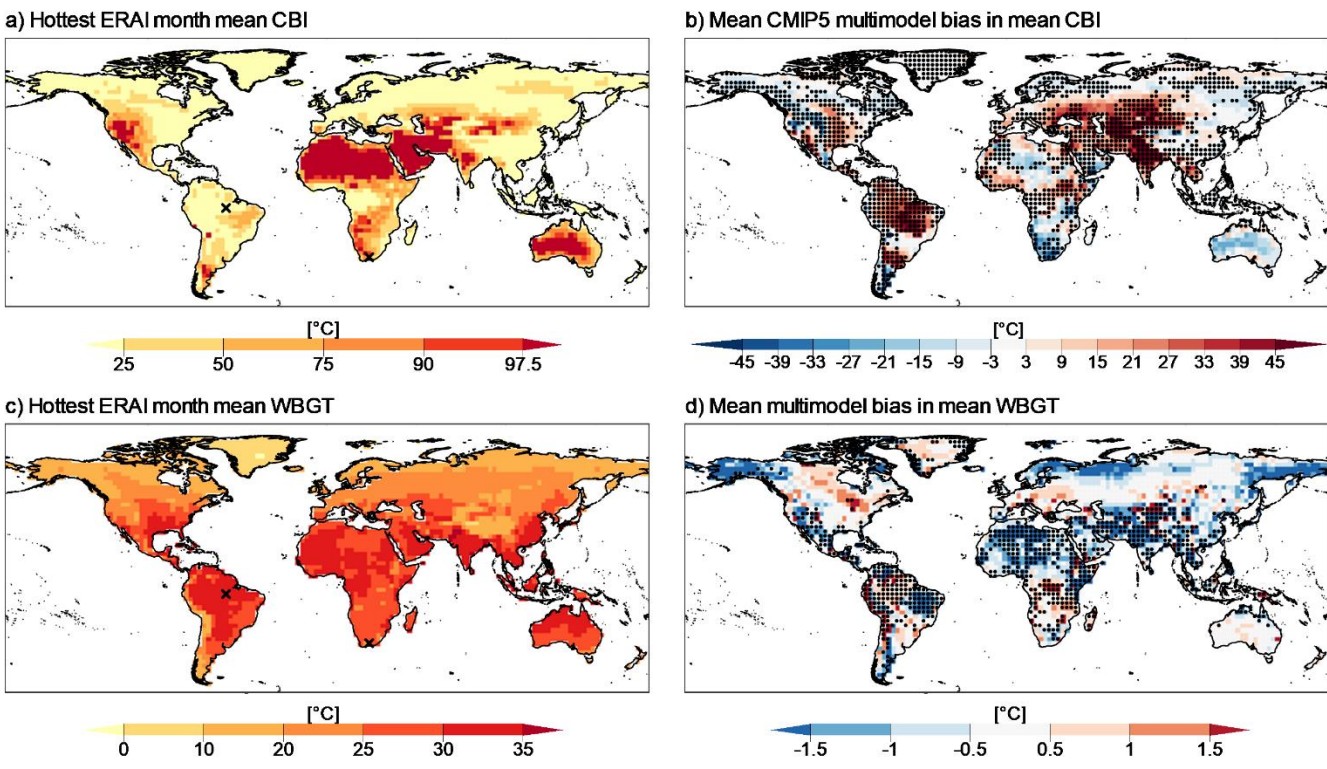

**Figure 3: Mean fire hazard index (*CBI*) value for ERA-Interim (a), mean multimodel bias in mean *CBI* (b). Note that the palette is nonlinear, as it follows typical defined ranges of fire hazard levels based on the *CBI*, i.e. Very Low, Low, Moderate, High, Very High, and Extreme. Mean heat stress index (*WBGT*) value for ERA-Interim (c), and mean multimodel bias in mean *WBGT* (d). Stippling indicates locations where at least 75% of CMIP5 models failed the AD two-sample test between the CMIP5 and ERA-Interim distributions of *CBI* and *WBGT*. Bias was calculated as (CMIP5 - ERA-Interim).**


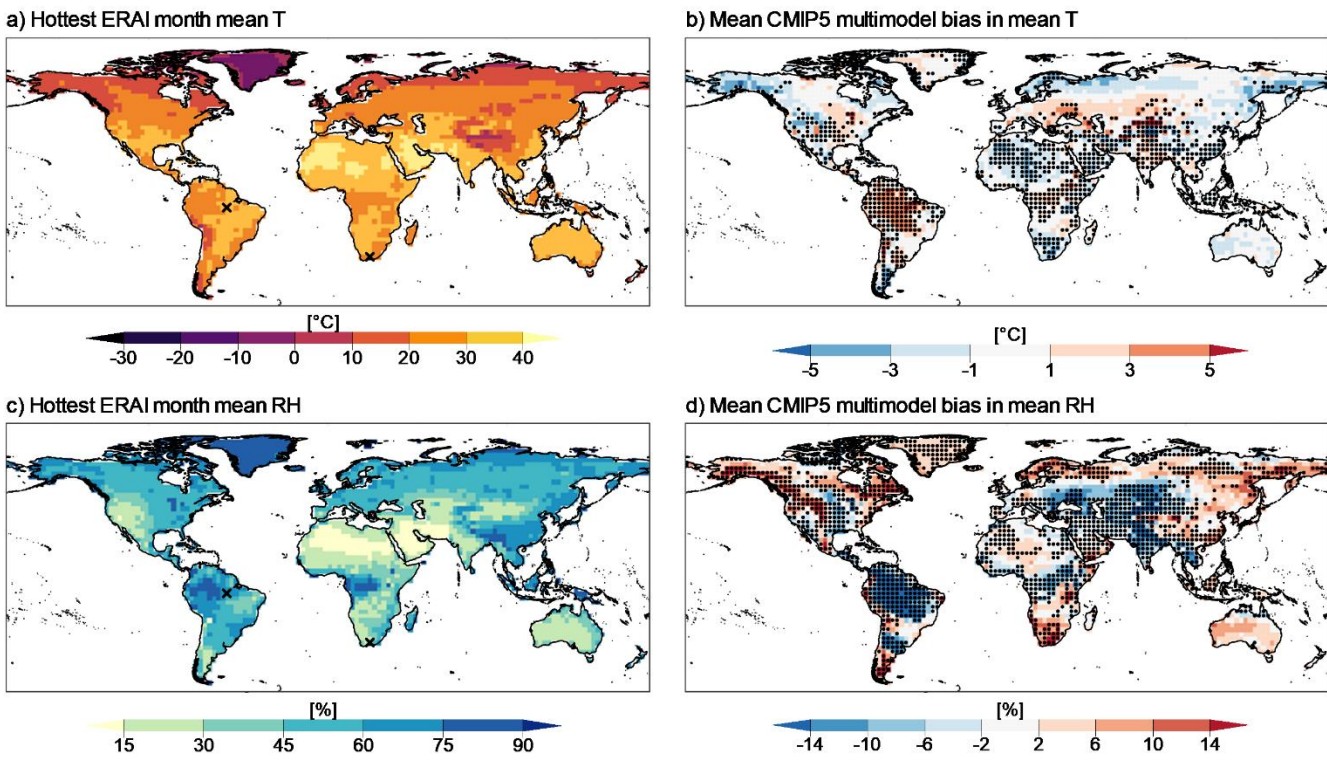


**Figure 4: Mean temperature (T) of the hottest month in ERA-Interim reanalysis (a), mean CMIP5 multimodel bias in mean temperature (b), mean relative humidity (RH) of ERA-Interim reanalysis (c), and mean CMIP5 multimodel bias in mean relative humidity (d). Stippling indicates locations where at least 75% of models failed the AD two-sample test between the CMIP5 and ERA-Interim marginal distributions of *T* and RH. Bias was calculated as (CMIP5 - ERA-Interim).**




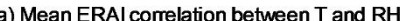

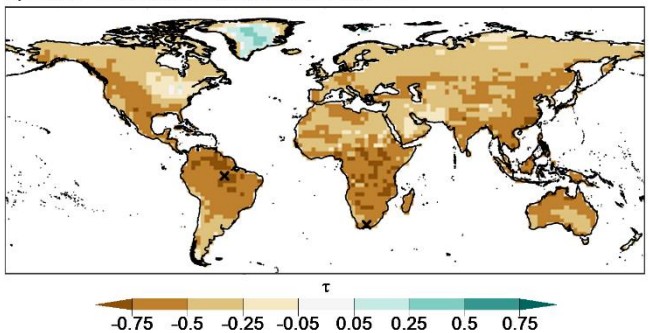

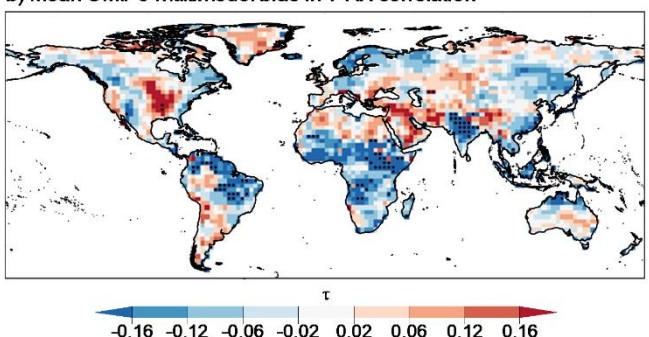

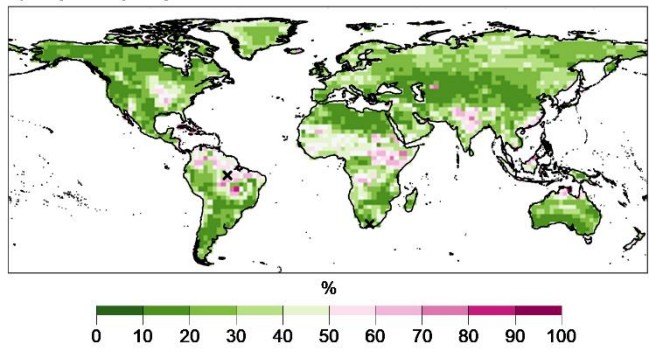

**Figure 5.** Mean ERA-Interim correlation (τ) between *T* and *RH* (a), mean CMIP5 multimodel bias in τ (b), the proportion of CMIP5 samples where the Copula equality was rejected (c). Stippling indicates locations where the correlation of more than 75% of CMIP5 model samples have significantly different values to ERA-Interim, as calculated using Bonferroni-corrected p-values. Bias was calculated as (CMIP5 - ERA-Interim).




**Figure 6: ERA-Interim (a) and CMIP5 multimodel mean (b) 95th quantile *CBI* values. Note that the palette is non-linear, as it follows typical defined ranges of fire hazard levels based on the *CBI*, i.e. Very Low, Low, Moderate, High, Very High, and Extreme. Mean CMIP5 multimodel bias in Q95 *CBI* (c), and its decomposition into bias due to the *T* (d), *RH* (e) and Copula (f) components of the model. Stippling indicates locations where more than 75% of CMIP5 model sample values lie outside the 95% confidence interval for ERA-Interim estimated based on bootstrap. Bias was calculated as (CMIP5 or Transformation - ERA-Interim).**




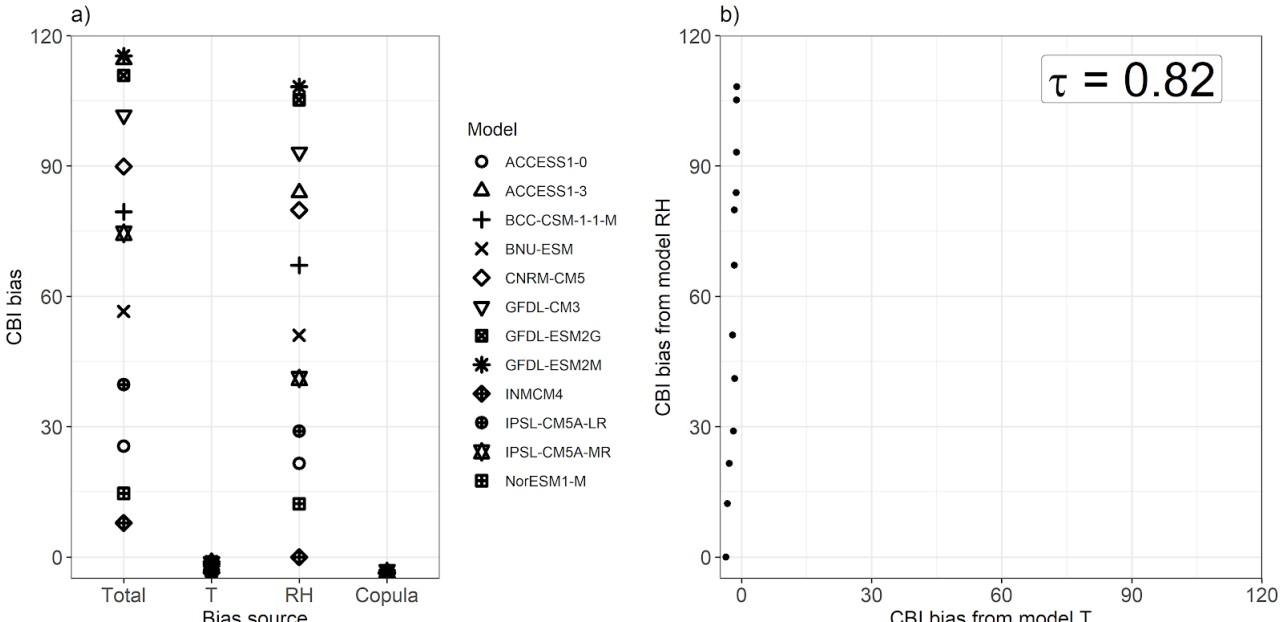

**Figure 7: Spread of mean total bias in the 95th quantile (Q95) of *CBI* and its contribution from T, *RH* and their copula for individual CMIP5 models (a), and a scatter plot of the *T* and *RH* contributions to Q95 *CBI* bias, with their Kendall rank correlation coefficient (p-value<0.001) (b). Shown are the results for a grid-point in Brazil (Amazonia, 5ºS and 56.5ºW). Bias was calculated as (CMIP5 or Transformation - ERA-Interim).**

**Figure 8: ERA-Interim (a) and CMIP multimodel mean (b) 95th quantile *CBI* values, mean CMIP5 multimodel bias in Q95 *CBI* (c), and its decomposition into bias due to the *T* (d), *RH* (e) and Copula (f) components of the model. Stippling indicates locations where more than 75% of CMIP5 model sample values lie outside the 95% confidence interval for ERA-Interim estimated based on bootstrap. Bias was calculated as (CMIP5 or Transformation - ERA-Interim).**


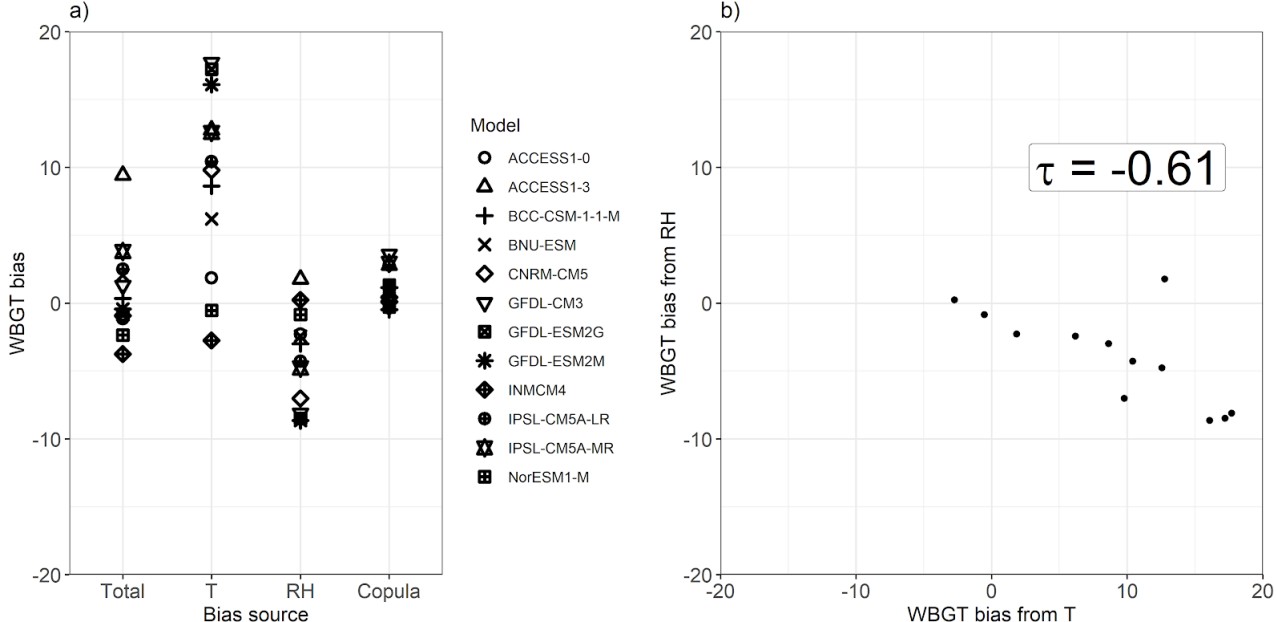

**Figure 9: Spread of mean total bias in the 95th quantile (Q95) of *WBGT* and its contribution from T, *RH* and their copula for individual CMIP5 models (a), and a scatter plot of the *T* and *RH* contributions to Q95 *WBGT* bias, with their respective Kendall rank correlation coefficient (p-value < 0.001). Shown are results for a grid-point in Brazil (Amazonia, 5ºS and 56.5ºW). Bias was calculated as (CMIP5 or Transformation - ERA-Interim).**






## Appendix A Information


**Figure A1: Samples of hourly 2-meter air Temperature (ºC) versus Relative Humidity (%) during the period 1979-2005 for ERA-Interim reanalysis (grey points) and 4 models (black points) from the CMIP5 multi-model ensemble (BNU-ESM (a), GFDL-CM3 CNRM-CM5 (b), GFDL-CM3 (c) and IPSL-CM5A-LR (d)) for a grid-point in Brazil (Amazon, 5ºS and 56.5ºW) indicated throughout map plots in Results section with X markers. The isolines illustrate equal levels of the hazard indices of fire (orange) and**
**heat stress (green), corresponding to *CBI* and *WBGT* indices, respectively, which are both functions of Temperature and Relative humidity.**



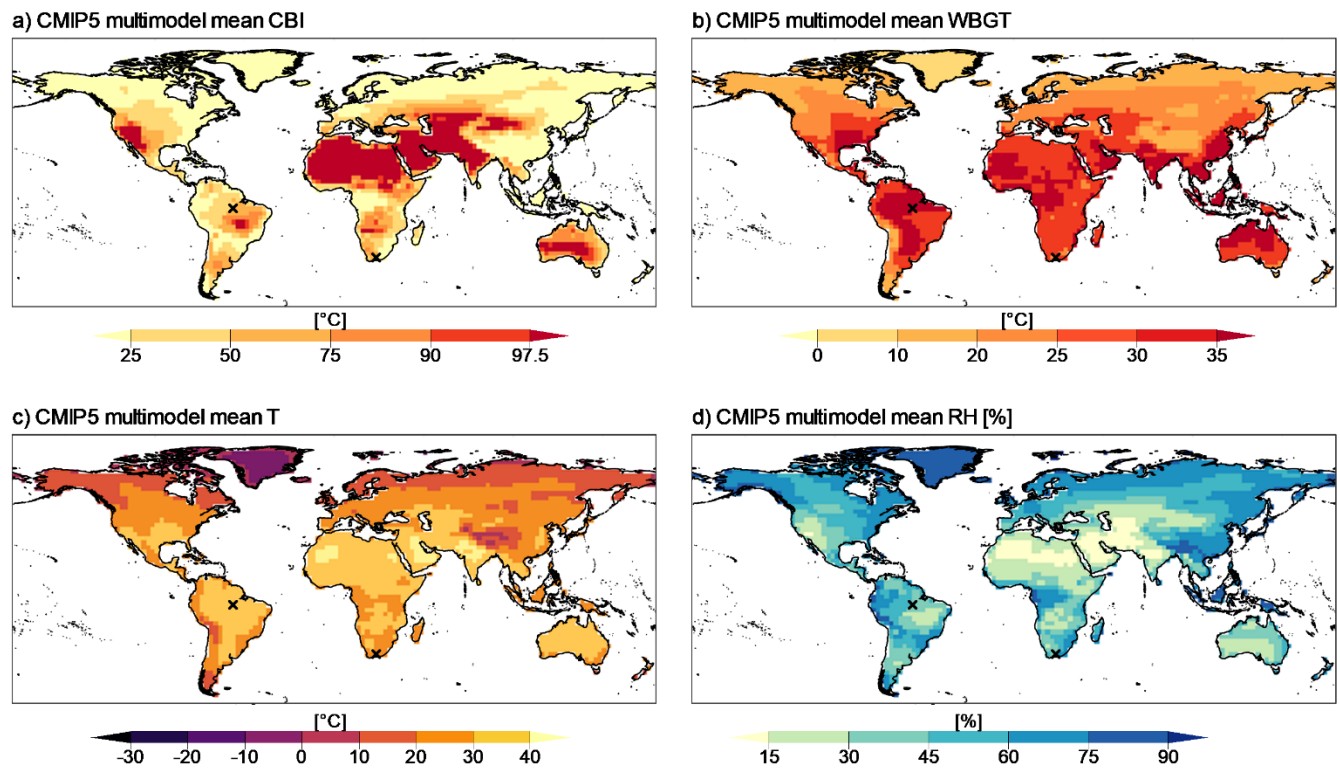

**Figure A2: CMIP5 multimodel mean fire hazard index (*CBI*) value (a), heat stress index (*WBGT*) value f (b), temperature (T) value (c), and mean relative humidity (RH) (d).**


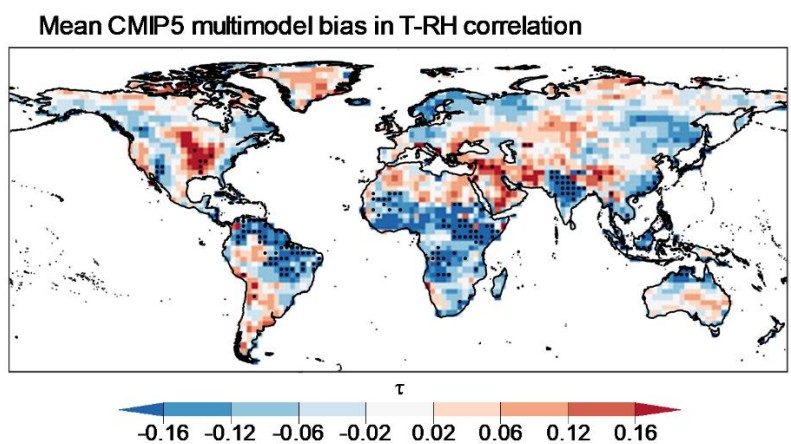

**Figure A3: As Figure 5b but where stippling indicates locations where more than 75% of CMIP5 model sample values lie outside the 95% confidence interval for ERA-Interim.**


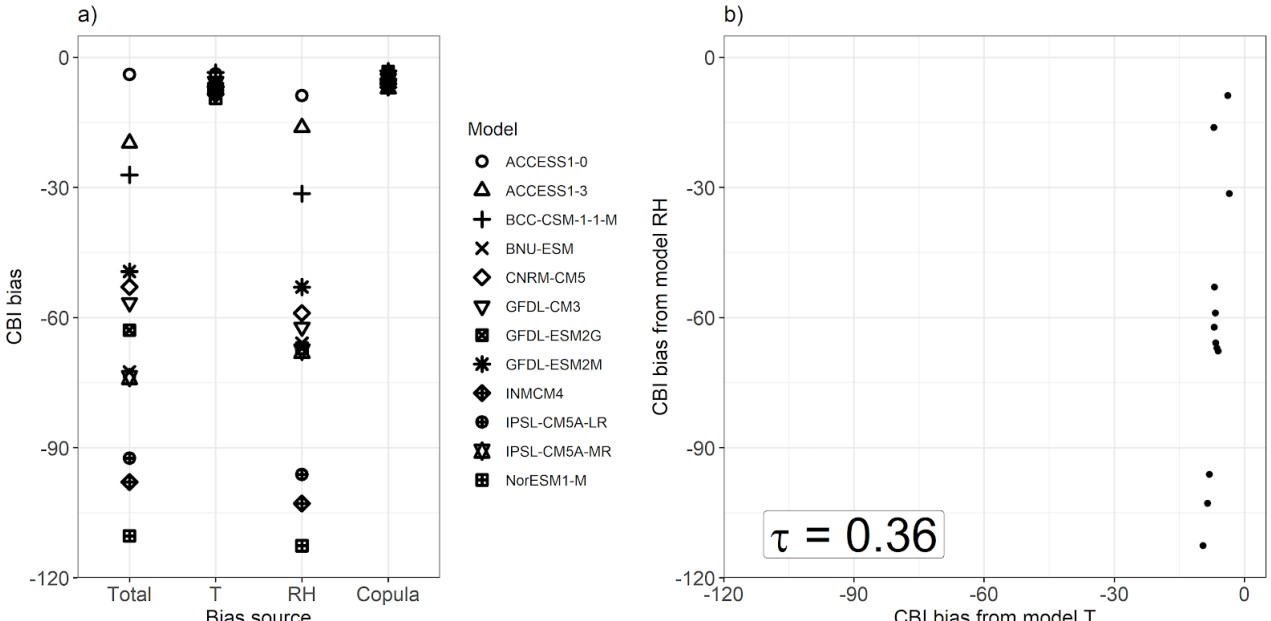

**Figure A4: As Figure 7 for a grid-point in South Africa (32.5ºS and 23.5ºE), Kendall rank correlation p-value = 0.12.**



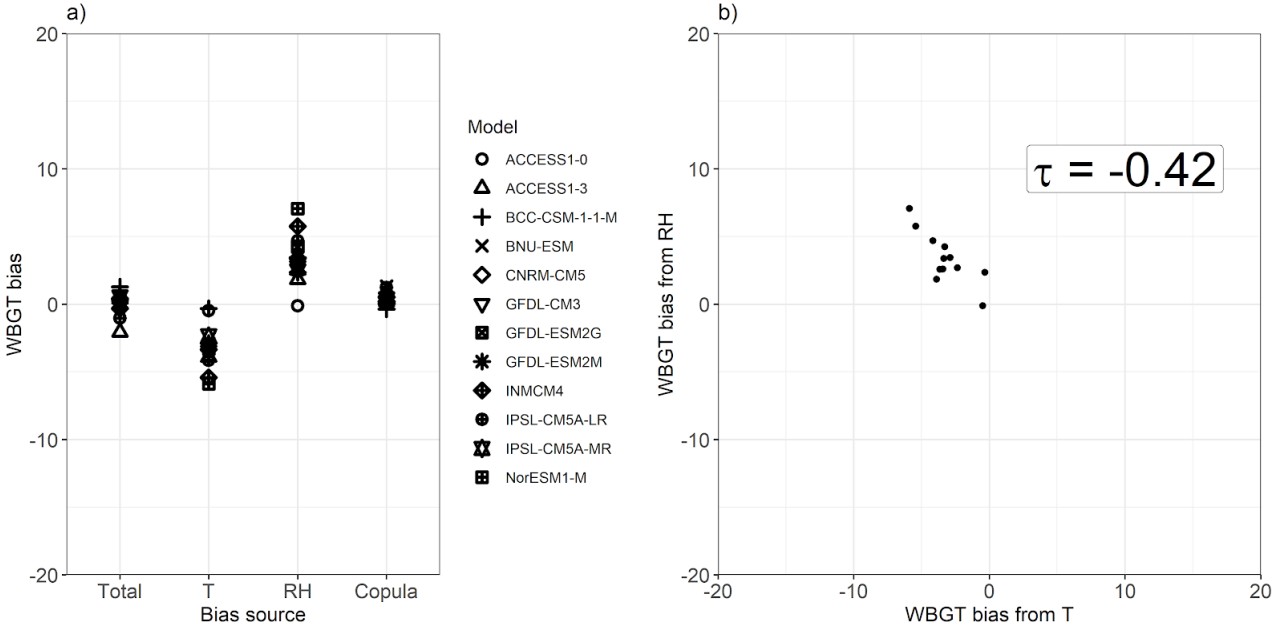

**Figure A5: As Figure 9 for a grid-point in South Africa (32.5ºS and 23.5ºE), Kendall rank correlation p-value = 0.063.**