# Peer review of "Towards a compound event-oriented climate model evaluation: A decomposition of the underlying biases in multivariate fire and heat stress hazards"

_Natural Hazards and Earth System Sciences, 2020_

## Referee Comment (RC1) · Anonymous Referee #1 · 7 Dec 2020

Relying on copula theory, the authors propose a multivariate bias assessment to separate biases in hazard from univariate drivers from their dependence. This is a relevant topic to understand compound events and how climate models can represent and simulate them or not. This framework is applied to two hazard indicators related to fire (Chandler Burning Index, CBI) and heat stress (wet-bulb globe temperature, WBGT) hazards. Overall, the paper is interesting and well constructed, with appropriate statistical experiments developed in Bevacqua et al., 2019. I enjoyed reading this submission. It clearly deserves to be published. Nevertheless, I have some comments that the authors may want to include in their manuscript. Those are mostly for clarifications and general consideration of the methodology.

Specific comments.

- L. 33-34: "well-designed physically based multivariate bias adjustment should be considered for hazards and impacts that depend on multiple drivers". I fully agree on this sentence that recurrently come back in the "bias correction" literature. However, it is not clear what this ("physically based adjustment)" means in practice, as no physically based bias adjustment is suggested (and we can argue that this is also the case in the literature). Do the authors have some in mind?

- Here, only inter-variable dependence (between T and RH) is considered. Can this framework be applied or extended to deal with temporal dependence (e.g., dependence between a variable and the same variable with a given lag) or spatial dependence? Or even both?

- In the same idea: here, each indicator is made only of 2 variables (temperature and relative humidity). Does the proposed framework work in 3 or 4 variables? I.e., in a higher dimensional context?

- In a context with more than two variables, I guess that the choice of the "non-parametric framework" (i.e., empirical distributions) is not appropriate anymore. This needs to be more discussed (although already briefly mentioned in the "discussion" section).

- Fig 1: the figure seems to show (visually) that the biases visible in panel c) mostly come from strong biases in the marginal (i.e., panels a and d) and not really from the dependence structure in panel b) that seems equivalent for ERAI and BNU. Is that correct? If so, I am not convinced that this model is the best example, as it would have been more informative/illustrative to show results for a model where both (marginal and dependence) contribute to the biases in the bivariate distribution.

[Figure]

- L.204-205 and 226-228: "if the model sample value of $\tau$ lies within the confidence interval calculated for its corresponding ERA-Interim sample, the model sample is judged to not significantly differ from ERA-Interim in terms of the rank correlation between T and RH." and "Like our evaluation of Kendall's $\tau$, if the model index lies outside the confidence interval we consider the model has a significantly different representation of extreme values of CBI and WBGT from ERA-Interim." This is a good approach that is accepted. However, one can wonder why not testing the other way around? (i.e., testing if ERAI lies in the interval from the model). Would this give equivalent results? Please, expand.

Minor/technical comments.

- L. 85: all analysEs

- L. 93-95: "we carry out the analysis on the de-correlated time series, which are obtained from the original through subsampling every N=9 days, this is the minimum lag required to remove the autocorrelation in T and RH time series data (at 95% confidence level)". Could the authors elaborate on this? E.g., how is "N=9 days" determined?

- L.115: As mentioned, "e" depends on T and RH but this link should be reminded in a few more details.

- L.200 and after: "z_{\alpha}" is not defined.

---

## Referee Comment (RC2) · Anonymous Referee #2 · 18 Jan 2021

The study by Villalobos-Herrera et al. deals with an important topic and offers an approach to identify and quantify the source of biases in multivariate impact-based indicators derived from climate model simulations. The methods are correct and well explained, although there are few clarifications needed (see the specific comments). Results are effectively presented and useful to start reflecting on this complex topic. Concerning the manuscript, I found the discussion session a repetition of concepts already discussed and again mentioned in the conclusions. I suggest to make it shorter and more focused on the key point, that to me is the difficulty in having multivariate bias

adjustment methods and the complexity of some impact indicators based on several different variables.

Specific Comments

Figure 1 caption is 'chaotic'. I suggest to re-write. Figure 2 is not very informative. I would either improve or remove. Figure 7 I would modify Panel b to let readers better appreciate the identified behaviour. L87: this implicitly means you assume models are able to correctly reproduce the seasonality. It may be worth to discuss it. L96 more details should be added on this estimated lag. L100-115 add a brief explanation on all these absolute numbers, just to let readers better understand. L157 following your notation Uerai is the transformed random variable (unif distributed) from Terai. L191-192 Since many tests exist to compare distributions, I do not understand this sentence on K-S and A-D. I would delete it. L213 Here, on the contrary, I would add an explanation. Why CvM test and not the A-D you use for marginals? As they both belong to the same test family. L319 According to the Figure, it seems that the dependence contributes much less than the others.

---

## Author Comment (AC1) · 24 Feb 2021

Reply to Reviewer #1 comments

We thank the Reviewer for the time they spent to review the manuscript. The helpful and constructive comments have contributed to improve the paper. The text below contains our responses to each comment.

General comments: Relying on copula theory, the authors propose a multivariate bias assessment to separate biases in hazard from univariate drivers from their depen-

[Figure]

dence. This is a relevant topic to understand compound events and how climate models can represent and simulate them or not. This framework is applied to two hazard indicators related to fire (Chandler Burning Index, CBI) and heat stress (wet-bulb globe temperature, WBGT) hazards. Overall, the paper is interesting and well constructed, with appropriate statistical experiments developed in Bevacqua et al., 2019. I enjoyed reading this submission. It clearly deserves to be published. Nevertheless, I have some comments that the authors may want to include in their manuscript. Those are mostly for clarifications and general consideration of the methodology.

We thank the Reviewer for the input and the positive feedback on the paper.

Specific comments:

C1: L. 33-34: "well-designed physically based multivariate bias adjustment should be considered for hazards and impacts that depend on multiple drivers". I fully agree on this sentence that recurrently come back in the "bias correction" literature. However, it is not clear what this ("physically based adjustment)" means in practice, as no physically based bias adjustment is suggested (and we can argue that this is also the case in the literature). Do the authors have some in mind?

With this phrase we meant to refer to the design of the overall procedure, also including the climate model selection, which is taken up in the discussion ("Climate model output should be a reliable input for the bias adjustment methods, e.g., models should provide a plausible representation of large-scale atmospheric circulation (Maraun et al., 2016; 2017).")

To make this clearer we will modify the abstract, using the word "procedure": "well-designed physically-based multivariate bias adjustment procedure"

And we will further modify the discussion, to explicitly refer to the above: "These findings exemplify the need for multivariate bias adjustment methods, which can adjust climate model biases in the dependencies between multiple drivers of hazards (Francois et al., 2020; Vrac, 2018). Furthermore, relying on climate models that plausibly represent large-scale atmospheric circulation (Maraun et al., 2016; 2017) would improve our confidence in the simulation of multivariate hazards."

C2-4: - Here, only inter-variable dependence (between T and RH) is considered. Can this framework be applied or extended to deal with temporal dependence (e.g., dependence between a variable and the same variable with a given lag) or spatial dependence? Or even both? - In the same idea: here, each indicator is made only of 2 variables (temperature and relative humidity). Does the proposed framework work in 3 or 4 variables? I.e., in a higher dimensional context? - In a context with more than two variables, I guess that the choice of the "nonparametric framework" (i.e., empirical distributions) is not appropriate anymore. This needs to be more discussed (although already briefly mentioned in the "discussion" section).

While the paper focussed on disentangling biases in individual physical drivers of compound events, it is in principle possible to extend the framework to higher dimensions. Cases with more than two driving variables require the use of multivariate copulas (e.g., vine copulas to decompose the dependencies between variables or other methods to build the multivariate model). The same applies for incorporating temporal dependencies. Overall, considering more dimensions adds complexity and would require larger sample sizes but does not limit the method to the bivariate case. To help clarify this we will add the following to the discussion at the end of L.384:

"... For example, in the case of three variables X1, X2, and X3, we would have to investigate the behaviour of marginals and then the dependence between X1 and X2 (with the 2-Copula C12), X2 and X3 (C23), and X1 and X3 (C13), and then joint behaviour of the three variables with the 3-Copula (C123). Similar considerations apply for the consideration of temporal dependencies. The analysis can be done using both a parametric or non-parametric approach. For instance, in Vezzoli et al. (2017), a non-parametric approach has been used to analyse the behaviour of the three variables precipitation, temperature and runoff."

The bias decomposition framework for each individual variable only relies on the empirical distributions of model and reference observations and thus does not change for cases with more than two variables.

C5: - Fig 1: the figure seems to show (visually) that the biases visible in panel c) mostly come from strong biases in the marginal (i.e., panels a and d) and not really from the dependence structure in panel b) that seems equivalent for ERAI and BNU. Is that correct? If so, I am not convinced that this model is the best example, as it would have been more informative/illustrative to show results for a model where both (marginal and dependence) contribute to the biases in the bivariate distribution.

Thank you very much for the suggestion. We agree that an illustrative example of the bias contribution from both the marginals and the copula would be more appropriate for this figure. In the revised manuscript, we will modify Figure 1 by replacing the model BNU-ESM with the model IPSL-CM5A-LR, which presents bias contributions from both the marginals and the copula (see proposed figure at the end and in the attached pdf). In addition, and in following with the comments by Reviewer 2, we will simplify the caption to:

Figure 1: Copula-based conceptual framework employed in this study to evaluate biases in CBI and WBGT indices. The framework is illustrated for a representative location in Brazil (Amazon, 5°S and 56.5°W; indicated via X markers in the next figures). Panel (c) shows the bivariate distribution of T and RH based on ERA-Interim (grey) and IPSL-CM5A-LR data (black) during 1979-2005. Isolines indicate equal levels of CBI (orange) and WBGT (green). The decomposition of biases from the marginals (a, d) and the copula (b) are illustrated as the discrepancies between the black (IPSL-CM5A-LR model) and grey features (ERA-Interim).

C6: - L.204-205 and 226-228: "if the model sample value of $\tau$ lies within the confidence interval calculated for its corresponding ERA-Interim sample, the model sample is judged to not significantly differ from ERA-Interim in terms of the rank correlation be-

tween T and RH." and "Like our evaluation of Kendall's $\tau$, if the model index lies outside the confidence interval we consider the model has a significantly different representation of extreme values of CBI and WBGT from ERA-Interim." This is a good approach that is accepted. However, one can wonder why not testing the other way around? (i.e., testing if ERAI lies in the interval from the model). Would this give equivalent results? Please, expand.

Thank you for this interesting question. Testing the other way around would be computationally much more expensive as confidence intervals would need to be calculated for every model, and in addition it would produce results which would be more difficult to interpret. We thus consider our approach preferable and all models are tested against the same reference.

MINOR COMMENTS:

- L. 85: all analysEs

Thank you, this has been corrected.

- L. 93-95: "we carry out the analysis on the de-correlated time series, which are obtained from the original through subsampling every N=9 days, this is the minimum lag required to remove the autocorrelation in T and RH time series data (at 95% confidence level)". Could the authors elaborate on this? E.g., how is "N=9 days" determined?

We agree that greater detail would be helpful here and will add the following details to the manuscript: "... we carry out the analysis on the de-correlated time series, which are obtained from the original through subsampling every N=9 days, where N is the lag required to remove the autocorrelation in T and RH time series data everywhere (at 95% confidence level). The value of N was determined as follows: for all grid points and years in ERA-Interim and the CMIP5 models, the autocorrelation function was calculated; then, the minimum lag for which the autocorrelation was non-significant at the 95% confidence level was determined. Finally, the maximum of all the minimum

lags was selected, resulting in N=9 days. The time series for all models and locations are sampled with the frequency of N. This is done N-times using different start epochs, where the first sampled time series starts with time epoch one, the second sampled time series with time epoch two and so on up to nine. The de-correlated time series of T and RH will henceforth be simply referred to as samples in the following sections."

- L.115: As mentioned, "e" depends on T and RH but this link should be reminded in a few more details.

We are not sure how to interpret this comment. Here we simply state what is evident from the equation used to compute e, namely that it depends on air temperature and relative humidity.

- L.200 and after: "z_{\alpha}" is not defined.

This is the quantile of the standard normal distribution; the following will be added to the paper: "where $\sigma^2$ is an estimator of var($\tau$) and z_($\alpha/2$) is the quantile of the standard normal distribution for $\alpha/2$ (Hollander et al., 2014)."

Please also note the supplement to this comment:
https://nhess.copernicus.org/preprints/nhess-2020-383/nhess-2020-383-AC1-supplement.pdf

———————————————————

[Figure]

Figure 1: Copula-based conceptual framework employed in this study to evaluate biases in CBI and WBGT indices. The framework is illustrated for a representative location in Brazil (Amazon, 5ºS and 56.5ºW; indicated via X markers in the next figures). Panel (c) shows the bivariate distribution of T and RH based on ERA-Interim (grey) and IPSL-CM5A-LR data (black) during 1979-2005. Isolines indicate equal levels of CBI (orange) and WBGT (green). The decomposition of biases from the marginals (a, d) and the copula (b) are illustrated as the discrepancies between the black (IPSL-CM5A-LR model) and grey features (ERA-Interim).

**Fig. 1.** Modified figure 1, see Reviewer 1 comment 5

---

## Author Comment (AC2) · 24 Feb 2021

Reply to Reviewer #2 comments

Many thanks to the Reviewer for their comments and the time and effort that required to prepare them. We believe your input will contribute to improving our paper. You may find our responses to your comments below.

General comments: The study by Villalobos-Herrera et al. deals with an important topic and offers an approach to identify and quantify the source of biases in multivari-

[Figure]

ate impact-based indicators derived from climate model simulations. The methods are correct and well explained, although there are few clarifications needed (see the specific comments). Results are effectively presented and useful to start reflecting on this complex topic. Concerning the manuscript, I found the discussion session a repetition of concepts already discussed and again mentioned in the conclusions. I suggest to make it shorter and more focused on the key point, that to me is the difficulty in having multivariate bias adjustment methods and the complexity of some impact indicators based on several different variables.

Thank you for your positive feedback. We have re-written a portion of our discussion, in particular L370-L384 to highlight the importance of multivariate adjustment methods and hazard indicator complexity. This will be implemented in the revised manuscript:

Our results underline the importance of attributing the sources behind biases in multivariate hazard indicators beyond simple univariate assessments. Biases in multivariate hazard indicators can be rather complex, as exemplified by our two example indicators which show very different bias structures despite being constructed by the same variables. Climate models that tend to simulate too high T also tend to simulate too low RH and vice versa (Fischer and Knutti, 2013), this behaviour would be expected to cause compensating biases in WBGT and enhance biases in CBI. In fact, biases in WBGT are smaller than the bias contributions from T and RH, demonstrating the presence of compensating biases for WBGT. A negative inter-model correlation between the contributions of T and RH to WBGT biases reduces the biases in WBGT in the CMIP5 average. While we have found a positive inter-model correlation between the bias driven by T and RH, no enhancement of the CBI bias occurs because CBI is mainly controlled by RH (see isolines in Figure 1c), which consequently also controls the bias of the index.

Specific comments: C1: Figure 1 caption is 'chaotic'. I suggest to re-write.

We have modified Figure 1 according to comments by Reviewer 1 (see figure at the

bottom and in the attached supplement). The caption has been modified to read:

Figure 1: Copula-based conceptual framework employed in this study to evaluate biases in CBI and WBGT indices. The framework is illustrated for a representative location in Brazil (Amazon, 5°S and 56.5°W; indicated via X markers in the next figures). Panel (c) shows the bivariate distribution of T and RH based on ERA-Interim (grey) and IPSL-CM5A-LR data (black) during 1979-2005. Isolines indicate equal levels of CBI (orange) and WBGT (green). The decomposition of biases from the marginals (a, d) and the copula (b) are illustrated as the discrepancies between the black (IPSL-CM5A-LR model) and grey features (ERA-Interim).

C2: Figure 2 is not very informative. I would either improve or remove.

We believe Figure 2 facilitates reader understanding of the data processing procedure, however we agree that it may be superfluous in the main text and will move it to the Appendix.

C3: Figure 7 I would modify Panel b to let readers better appreciate the identified behaviour.

Figure 7b has equal axes for both sources of CBI bias (as does Figure 9b). We opted for this choice such to highlight the contrast between the small spread and magnitude of the T bias contribution relative to that of RH. We believe that having axes with different ranges would make this less evident. To help clarify this aspect we will add the following sentence to the figure caption:

Equal axes are used in (b) to highlight the differences in spread between both bias components.

C4: L87: this implicitly means you assume models are able to correctly reproduce the seasonality. It may be worth to discuss it.

Thank you for this suggestion, we agree and will add the phrase to our text:

"Following Zscheischler et al. (2019), we restrict our analysis to the hottest calendar month of the year, which is selected based on the climatology of ERA-Interim data at each grid point. This choice was made because arguably heat stress and fire hazards tend to be more frequent during the warmest period of the year, and it avoids dealing with seasonality, however we note that this assumes that CMIP5 models correctly reproduce the seasonality observed in ERA-Interim."

C5: L96 more details should be added on this estimated lag.

Thank you, we agree, the explanation will be expanded as shown below:

". . . we carry out the analysis on the de-correlated time series, which are obtained from the original through subsampling every N=9 days, where N is the lag required to remove the autocorrelation in T and RH time series data everywhere (at 95% confidence level). The value of N was determined as follows: for all grid points and years in ERA-Interim and the CMIP5 models, the autocorrelation function was calculated; then, the minimum lag for which the autocorrelation was non-significant at the 95% confidence level was determined. Finally, the maximum of all the minimum lags was selected, resulting in N=9 days. The time series for all models and locations are sampled with the frequency of N. This is done N-times using different start epochs, where the first sampled time series starts with time epoch one, the second sampled time series with time epoch two and so on up to nine. The de-correlated time series of T and RH will henceforth be simply referred to as samples in the following sections."

C6: L100-115 add a brief explanation on all these absolute numbers, just to let readers better understand.

Both the employed indices are widely used in the scientific literature. The parameters were defined within the original sources. Given the comment of the referee, we will add a sentence at the end of the section: ". . . More details on the definitions of the CBI and WBGT are available at McCutchan and Main ,1989; and ACSM ,1984"

C7: L157 following your notation Uerai is the transformed random variable (unif distributed) from Terai.

This is correct, we will clarify this in the text: "From the variable Terai we calculated the uniformly distributed transformed random variable UT,erai =FT, erai(Terai)."

C8: L191-192 Since many tests exist to compare distributions, I do not understand this sentence on K-S and A-D. I would delete it.

We agree and will remove this sentence.

C9: L213 Here, on the contrary, I would add an explanation. Why CvM test and not the A-D you use for marginals? As they both belong to the same test family.

Thank you for the interesting question. Our goal is here to test whether two bivariate empirical copulas are equal. Hence, we use the test by Remillard and Scaillet (2009), which was specifically developed for this task. In general, we note that if we used a multivariate version of the A-D test and the null hypothesis (of equality in distribution) was rejected, then we would have no way of knowing whether the difference in the bivariate distributions was due to a difference in the marginal distributions or a difference in the dependence structure (i.e. the copula) or both.

To clarify why we use this test will add some text to the paper (in square brackets below):

"Note that different copulas may give rise to the same value of $\tau$, therefore we cannot conclude that a model that faithfully reproduces the ERA-Interim values of $\tau$ is accurately representing the full dependence structure between T and RH. Therefore, we account for [differences] in the dependence structure by also carrying out hypothesis tests which are based on the full copula function. We perform the non-parametric test of copula equality based on the Cramer-von-Mises test statistic proposed by Remillard and Scaillet (2009)," used in Vezzoli et al. (2017) for testing the capability of a climate-hydrology model to reproduce the dependence between temperature, precipitation and discharge for the Po river basin in Italy, and recently employed by Zscheischler and Fischer (2020) for evaluating the ability of climate models to represent the dependence between temperature and precipitation in Germany. The copula equality test has a null hypothesis of H0: Cerai = Cmod where Cerai and Cmod are the copulas of T and RH represented in ERA-Interim and a given model respectively, with the alternative hypothesis being that these copulas differ. [Unlike the AD test, which can evaluate CMIP5 model performance in reproducing a single marginal distribution, the copula equality test was specifically developed to test whether two empirical copulas are equal and thus evaluates the capacity of models to reproduce the full dependency structure between T and RH.] We used the TwoCop function of the TwoCop R-package (v1.0, Remillard and Plante, 2012) to run the test."

C10: L319 According to the Figure, it seems that the dependence contributes much less than the others.

We agree, as we discuss later in the paragraph. We open the paragraph at line 319 with an introductory sentence where we do not give details on the contributions of the copula and the two marginals. We feel this is warranted as the contribution of the dependence varies in space. This is discussed, together with the comment of the referee, at L330:

"In addition, we observe a tendency towards a lower bias, on average, driven by the copula component (global area weighted average of absolute bias equal to 0.85°C); note that, however, some relevant positive bias contributions exist over eastern Brazil and central Africa where the copula test shows higher frequencies of rejection (Figure 5c), and negative contributions over northern Russia, Central United States, and eastern Europe (Figure 8f). "

Please also note the supplement to this comment:
https://nhess.copernicus.org/preprints/nhess-2020-383/nhess-2020-383-AC2-supplement.pdf

[Figure]

Figure 1: Copula-based conceptual framework employed in this study to evaluate biases in CBI and WBGT indices. The framework is illustrated for a representative location in Brazil (Amazon, 5ºS and 56.5ºW; indicated via X markers in the next figures). Panel (c) shows the bivariate distribution of T and RH based on ERA-Interim (grey) and IPSL-CM5A-LR data (black) during 1979-2005. Isolines indicate equal levels of CBI (orange) and WBGT (green). The decomposition of biases from the marginals (a, d) and the copula (b) are illustrated as the discrepancies between the black (IPSL-CM5A-LR model) and grey features (ERA-Interim).

**Fig. 1.** Modified figure 1